



# High Reynolds number wind turbine blade equipped with root spoilers. Part II: Impact on energy production and turbine lifetime

Thomas Potentier[1,3], Emmanuel Guilmineau[1], Arthur Finez[2], Colin Le Bourdat[3], and Caroline Braud[1]

[1]LHEEA (Centrale Nantes / CNRS), 1 rue de la Noë 44321 Nantes Cedex 3 - FRANCE
[2]ENGIE Green, 59 Rue Denuzière 69002 Lyon - FRANCE
[3]ENGIE Green, 15 rue Nina Simone 44000 Nantes - FRANCE

**Correspondence:** Thomas Potentier (thomas.potentier@ec-nantes.fr)

**Abstract.** A wind turbine blade equipped with root spoilers is analysed using time domain aeroelastic Blade Element Momentum (BEM) simulations to assess the impact of passive devices on the turbine Annual Energy Production (AEP) and lifetime. Previous 2D Computational Fluid Dynamics (CFD) showed a large unsteadiness in aerodynamic coefficients associated to the spoiler, such behaviour is captured by the OpenFAST simulations when all degrees of freedom are switched off. Once the

turbine is fully flexible, a novel way to account for aerofoil generated unsteadiness in the fatigue calculation is proposed and detailed. The outcome shows that spoilers, on average, can increase the AEP of the turbine. However, the structural impacts on the turbine can be severe if not accounted for initially in the turbine design.

## 1 Introduction

Thanks to a steady rotor size increase over the last decades, the wind energy sector managed to grow. In the onshore wind

sector, due to various limitations, the rotor diameter remains constrained but blades over 60m long are now common. Larger blades requires more attention to details during the design phase to reduce the cost. The maintenance cost during the turbine lifetime increases too, a good understanding of the turbine ageing is necessary.

In order to reduce the Levelised Cost of Energy (LCOE), turbine manufacturers had to imagine solutions to increase the energy output of existing turbines. Among such solutions, there are Aerodynamic Add-On (AAO) which are mostly passive devices

attached onto the blade surface to either lower the acoustic emission or increase locally the power extraction.

With the increasing rotor diameter and hub height, turbine manufacturers are now facing aeroelastic challenges where tower and blades can deform over large distances. Before several extensive measurement campaigns of scaled models in large wind tunnels or in the field were performed (see Hand et al. (2001); Simms et al. (2001); Snel et al. (2007); Boorsma and Schepers (2014); Madsen et al. (2010); Trodborg et al. (2013)), the physical phenomena of wind loading unsteadiness was poorly

understood and large safety factors were used to ensure the turbine robustness and design lifetime. High fidelity tools could perform that task such as Computational Fluid Dynamics (CFD) but the computational cost would render the turbine's time to market too important. In another hand, turbine designers using quicker engineering tools such as Blade Element Momentum (BEM) lacked, at first, the necessary unsteady models. Now, it is common knowledge to be using unsteady models to simulate wind turbines and it is referenced in many textbooks (Hansen (2015); Burton (2001)). The large scale unsteadiness investigated



showed that the atmospheric boundary layer, turbulent wind, yawed inflow or even blade pitching can have serious impact on
the turbine if not properly accounted for, as found in Potentier et al. (2021a). One of the remaining challenge to predict even
more accurately the turbine loads is to account for the local unsteadiness, self generated by the flow travelling around an
aerofoil, that can be interacting with large scale unsteadiness. As detailed in Potentier et al. (2021b), where thick blade profiles
equipped with blade root AAO were studied with 2D CFD with low turbulence intensity free stream condition; the spoiler
produces the desired higher lift by reorganising the flow and pressure distribution around the aerofoil. In consequence the
unsteady behaviour (vortex shedding) behind the aerofoil is vastly different from the bare blade.

The aim of this paper is to investigate the effect of local unsteadiness, caused by the spoiler, at turbine level in terms of lifetime
impact. Because performing fatigue calculation using CFD would be too computationally expensive, and BEM cannot account
for scales of unsteadiness, we propose in this paper to bridge the gap by utilising the strengths of both simulation methods.
First the methodology to build the aeroelastic model is explained in Section 2. The use of 2D CFD associated with aeroelastic
BEM simulation will allow us to compare between the two configurations, the aerodynamic parameters such as lift coefficient
($C_L$), rotor loads, power and energy production (Section 3.1). A novel way of accounting for polar unsteadiness in the fatigue
lifetime calculation is proposed in Section 3.2.

## 2 Methodology

### 2.1 Wind turbine blade and aerofoil shape

The wind turbine geometry used in the present study was acquired during a scanning campaign on an operating 2MW turbine
(see Dambrine (2010)). During the scan post-treatment the chord, twist and thickness were also extracted, defining the blade
geometry (see Figure 1). One can note the typical "de-twist" toward the blade tip to alleviate the blade loading. The blade
geometry is discretised more densely at the root of the blade since the spoiler is installed at this location. More details about
the scan post-treatment are available in Potentier et al. (2021b). The scanned blade was originally equipped with root spoilers.
The blade without spoiler was generated by manually removing parts of the cloud points corresponding to the spoiler location,
consequently wherever the spoiler is not present both aerofoils geometries are identical. For the rest of the study, the simulations
will take place on the real scale, i.e. using the scan outputs as blade geometry.

### 2.2 Unsteady aerodynamic BEM inputs

The tools used to perform the spoiler impact assessment are: CFD for the polars generation and Blade Element Momentum
(BEM) theory for the aerodynamic calculations. BEM is used to calculate associated loads and compute the turbine Annual
Energy Production (AEP). The BEM solver used is the AeroDyn module (see Jonkman et al. (2015)) from OpenFAST[1].
OpenFAST can produce a large variety of *sensors* which are calculated outputs during the simulation.
AeroDyn is a well known tool developed by NREL, and has been used in many international or academic projects. A thorough

---

[1]https://openfast.readthedocs.io/en/main/ website accessed 08/11/2021

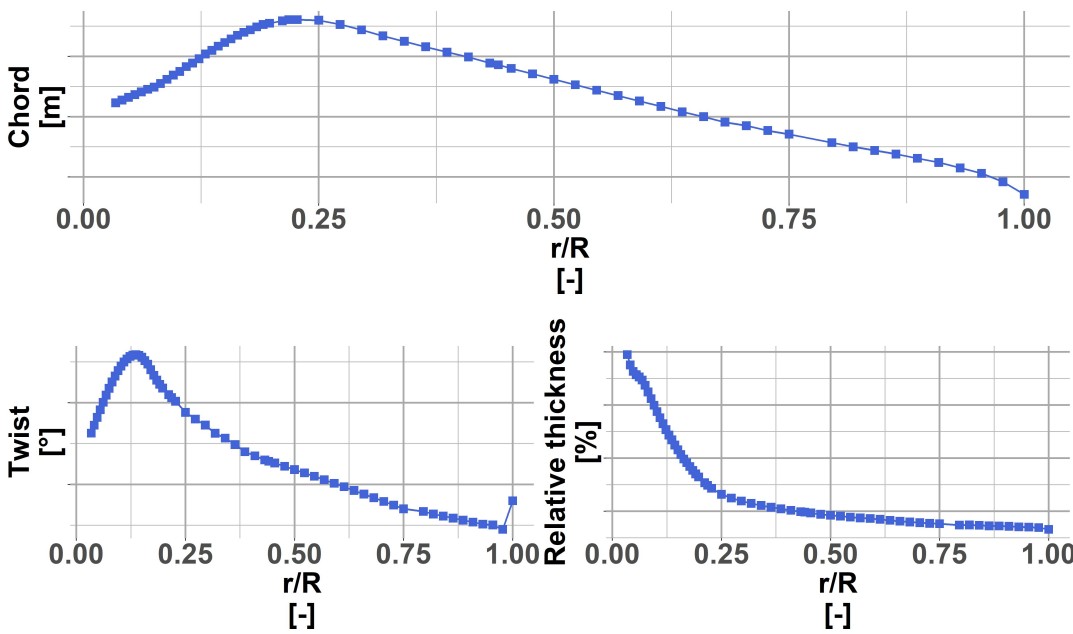

**Figure 1.** The scanned blade geometry: chord, twist and relative thickness distribution against the normalised radius.

explanation of the BEM theory is available in textbooks such as Hansen (2015) or Burton (2001). A brief step-by-step approach written below summarises the iterative BEM procedure:

1. The axial induction factor $a$, and the tangential induction factor $a'$, are first estimated (typically $a = a' = 0$).

2. Then, the inflow angle $\varphi$ is estimated from the instantaneous velocity inflow, $V_w$, the rotor rotational speed $\omega$, and the local radius $r$

3. The angle of attack, $\alpha$, is computed using the Blade Element Theory (BET) with $\theta$ the local twist angle and $\beta$ the blade pitch angle

4. Read the $C_L, C_D$ (drag coefficient) and $C_M$ (moment coefficient around the 1/4 chord position) from the polar associated to the analysed radius

5. Calculate the loads in the rotor plane using $C_L$, $C_D$ and $\varphi$

6. To account for the finite blade span, the Prandtl's tip correction factor is calculated

7. The initial induction coefficients, $a$ and $a'$, are updated accounting for highly loaded rotors

8. The unsteady BEM equations can be applied: yaw models, dynamic wake model, blade acceleration due to its deflection and tower shadow effect.



9. A convergence criteria, $\epsilon$, is defined and the iteration process restarts from step 2 until the convergence criteria is reached.

10. After convergence, the local loads (aerofoil level) can be calculated

11. Once all elements are converged, the integrated loads (rotor and turbine level) can be computed

The procedure described relies on steady polar to perform the iterative steps, it is an inherent BEM limitation. However, as highlighted in Potentier et al. (2021b), aerodynamic properties becomes highly unsteady at the blade root when a spoiler is present. To overcome the single steady polar limitation and use the unsteady coefficient, we decided to generate three steady

polars corresponding to the mean, minimum, and maximum $C_L$, $C_D$ and $C_M$ for each turbine case: "no spoiler" and "spoiler" (see Table 1). Those mean, minimum and maximum coefficients are representative of the states reached by the aerodynamic coefficients during the time series calculated using 2D CFD as found in Potentier et al. (2021b) (see e.g. Figure 2).

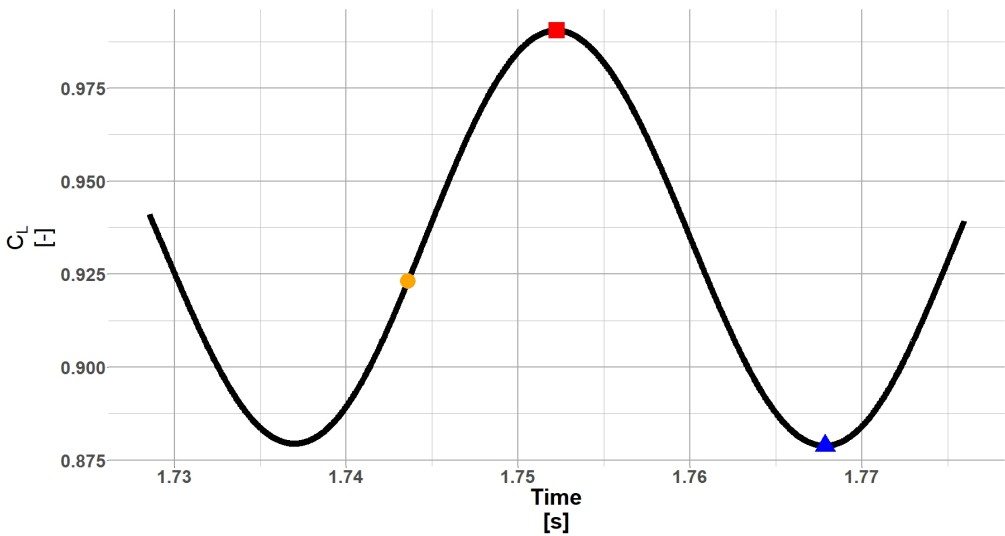

**Figure 2.** "Spoiler" case $C_L$ evolution in time $\alpha = 6°$ and $Re_c = 3 \times 10^6$. The blue triangle (▲) corresponds to the minimum $C_L$, the red square (■) corresponds to the maximum $C_L$, the orange dot (●) corresponds to the mean $C_L$.



**Table 1.** Turbine configurations analysed

| Spoiler configuration | Aerodynamic coefficients values |
|:---:|:---:|
| No spoiler | Mean aerodynamic coefficients |
| No spoiler | Maximum aerodynamic coefficients |
| No spoiler | Minimum aerodynamic coefficients |
| Spoiler | Mean aerodynamic coefficients |
| Spoiler | Maximum aerodynamic coefficients |
| Spoiler | Minimum aerodynamic coefficients |



## 2.3 Turbine structure scaling

The scan does not give any information on the blade's material, since only the outer skin was measured. Material properties
is a crucial element for turbine design and as part of academic or wind turbine exploiting party, we do not have access to this
information. Therefore, for the rest of the aeroelastic study, the blade and tower mechanical properties will be scaled using the
open source NREL 5MW turbine (see Jonkman et al. (2009)). Some hypotheses and assumptions had to be made and will be
explained below.

Usually, scaling is made to reach the same level of stress or reach similarity in physical phenomena: Mach number, Reynolds
number, Froude number (see Campagnolo (2013)). Here, since the stress target values are unknown and the physics similarity
is already achieved: Mach, Reynolds and Froude number close enough between the NREL turbine and the ENGIE Green
turbine, we decided to scale the turbine based on geometric properties. The NREL turbine has a 63m long blade and its tower
is 87.6m high, in comparison the ENGIE Green turbine has a blade length of 45m and the tower height is 80m. Several scaling
procedure exist and have been described in Loth et al. (2017); Canet et al. (2020). The authors aimed at creating a sub-scale
model for wind tunnel or field testing, where the difference between both model is large (reduction factor up to 90). In our
case, we desired to use known mechanical properties and adjust them based on the smaller blade and tower length, so that the
ENGIE Green turbine behaves similarly to the NREL one. Therefore, the method used varies slightly compared to the literature
and is described below.

The blade structural properties needed are the edgewise and flapwise local stiffnesses along the radius: $EI_{xx}$ and $EI_{yy}$, as
well as the linear mass $M_L$. $E$ is the Young's modulus while $I_{xx}$ and $I_{yy}$ are respectively the in-plane and out-of-plane the
sectional moment of inertia. Assuming identical material is used to manufacture both blade, only the sectional inertiae $I_{xx}$ and
$I_{yy}$ vary. Since the sectional inertia varies based on geometric properties we decided to use the chord as the main driver for
the change in properties. The thickness could also have been chosen, but the chord was preferred because of its larger absolute
value. In our geometric scaling we multiply the NREL 5MW stiffnesses $EI_{xx}$ and $EI_{yy}$ by the ratio of the local chord along
the radius to the power four (see equation 1), thanks to dimensional analysis. Following the same reasoning, we assume an
identical material, the NREL 5MW linear mass needs to be multiplied by the chord ratio at the power one (see equation 2).
The same methodology is applied to the tower stiffnesses and mass.

$$EI_{xx_j}^{EG} = EI_{xx_j}^{NREL} \times \left( \frac{c_{r/R_j}^{NREL}}{c_{r/R_j}^{EG}} \right)^4 \qquad\qquad EI_{yy_j}^{EG} = EI_{yy_j}^{NREL} \times \left( \frac{c_{r/R_j}^{NREL}}{c_{r/R_j}^{EG}} \right)^4 \qquad\qquad (1)$$

Where $E$ is the material Young's modulus, $I_{xx}^{EG}$ and $I_{yy}^{EG}$ are the ENGIE Green's blade local inertiae, $I_{xx}^{NREL}$ and $I_{yy}^{NREL}$ are
the NREL's 5MW turbine initial local inertiae. $c_{r/R}^{NREL}$, $c_{r/R}^{EG}$ are the NREL's 5MW and ENGIE Green's blade chords at the
same spanwise location and the subscript $j$ shows the analysed station.

$$M_{L_j}^{EG} = M_{L_j}^{NREL} \times \left( \frac{c_{r/R_j}^{NREL}}{c_{r/R_j}^{EG}} \right) \qquad\qquad (2)$$





Where $M_{L_j}^{EG}$ and $M_{L_j}^{NREL}$ are the linear mass of both blades and the subscript $j$ shows the analysed station.

Moreover, the blade and tower modal shapes, necessary OpenFAST inputs, have been recalculated using the scaled mechanical properties. A Campbell diagram illustrates that despite the difference in length and mass, both turbines behave similarly, as desired (see Figure 3). All ENGIE Green's blade modes follow the NREL baseline turbine trend with a little offset due to the shorter blade. Regarding the tower, the first modes (fore-aft and side-to-side) are identical between both turbines, only the seconds modes show a clear offset towards the highest frequencies.

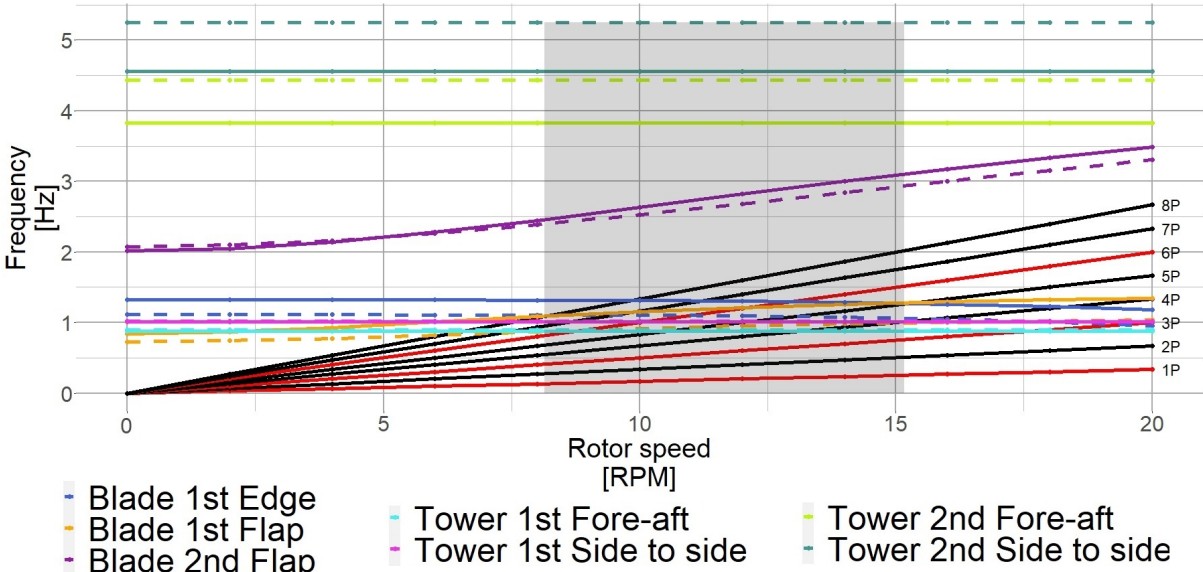

**Figure 3.** Campbell diagram comparison between the NREL reference turbine and the ENGIE Green scaled one. The solid lines shows the NREL response and the dashed lines the ENGIE Green turbine's response. The dark shaded area illustrates the ENGIE Green's turbine range of operation.

A final sanity check was performed on the mass to assess the validity of the scaling. The blade and tower mass where respectively 0.6% and 1.3% off compared to the manufacturer's design specifications, which is small enough to be acceptable. The Figure 4 and Figure 5 show the mechanical properties comparison with the original NREL 5MW turbine. Finally, the turbine characteristics publicly available and necessary to OpenFAST are gathered (see Appendix A).

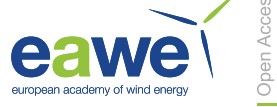


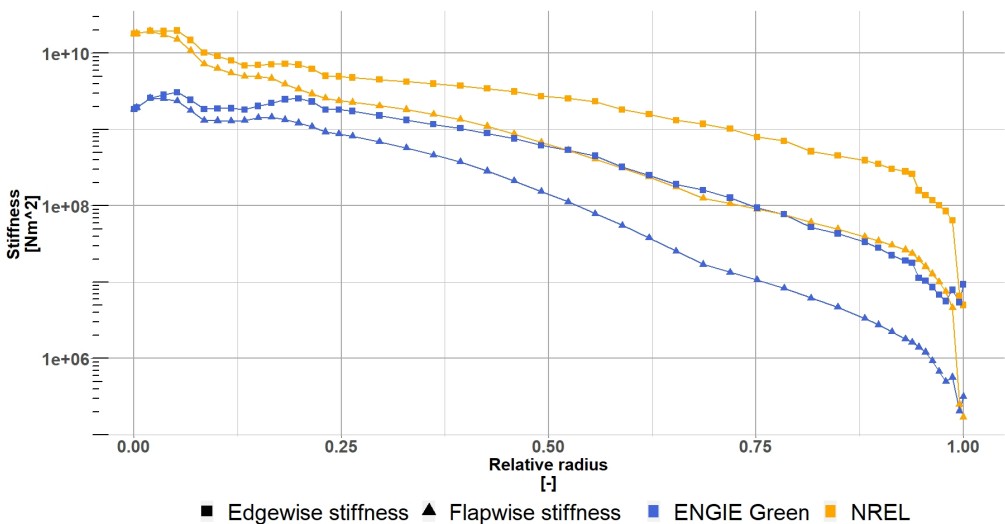

**Figure 4.** Blade stiffness properties. The blue lines (—) shows the scaled blade and the orange lines (—) the original NREL 5MW blade properties. The symbols ■ and ▲ show the different stiffness directions.

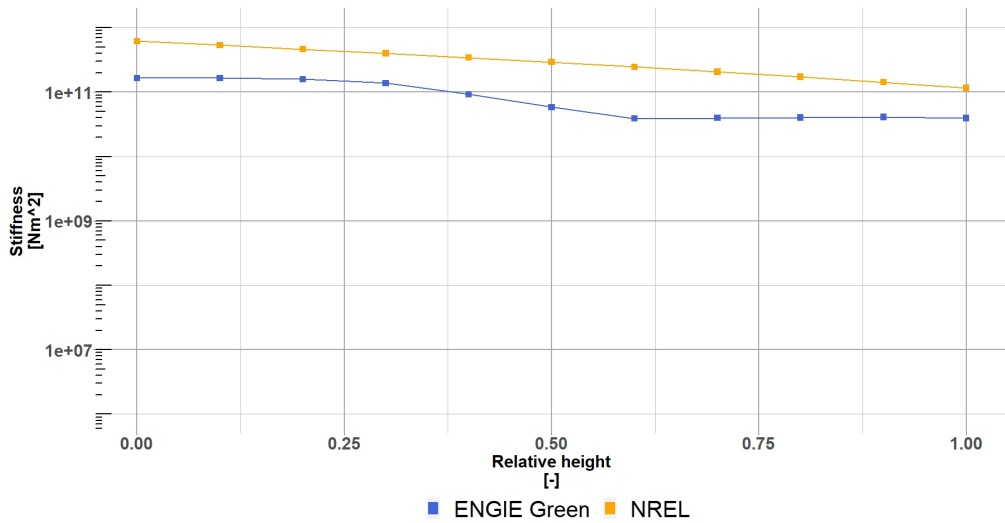

**Figure 5.** Tower stiffness properties. The blue lines (—) shows the scaled tower and the orange lines (—) the original NREL 5MW tower properties. Because fore-aft and side-to-side stiffnesses are identical only a single curve per tower is shown.



## 2.4 Unsteady polars generation

The grid independence study and polar generation methodology have already been performed and presented in Potentier et al. (2021b). Then, all 16 profiles listed in the Table 2 were computed to extract aerofoils aerodynamic coefficients (lift, drag and moment coefficients) for OpenFAST computations. Thus producing six different steady polars for the turbine (see Table 1). The Figure 6 and Figure 7 show representative sections for the lift and drag coefficient along the blade span. The solid lines show the mean aerodynamic coefficients values, while the shaded areas illustrates the range of variation reached during each

angle of attack calculation. Consequently, the polar using the maximum aerodynamic coefficients corresponds to the upper limit and the the polar using the mininum aerodynamic coefficient follows the lower limit.

Initial BEM simulations showed that high angles of attack can be reached ($\alpha > 50°$) for the inner sections, for this reason the inboard sections polars have been simulated up to $\alpha = 60°$. Each polar has then been extrapolated using the Viterna extrapolation method from Viterna and Janetzke (1982) to cover the full 360° range (-180 $\leq \alpha \leq$ 180). Then, to account for

the rotational effects, the 3D correction model derived by Chaviaropoulos was used (see Chaviaropoulos and Hansen (2000)).

**Table 2.** CFD calculated blade sections polars defining the BEM model assuming an inflow between 8m/s and 8.5m/s

| Spanwise location from the hub [m] | Relative spanwise location from the hub [%] | Aerofoil relative thickness [%] | Local Reynolds number [-] |
|---|---|---|---|
| 2.1 | 4.4% | 93% | $1.35 \times 10^6$ |
| 3.0 | 6.7% | 86.8% | $1.53 \times 10^6$ |
| 3.6 | 8.0% | 81.2% | $1.67 \times 10^6$ |
| 4.2 | 9.3% | 74.9% | $1.86 \times 10^6$ |
| 4.5 | 10.0% | 71.9% | $1.95 \times 10^6$ |
| 5.1 | 11.3% | 66.2% | $2.17 \times 10^6$ |
| 5.4 | 12.0% | 63.4% | $2.29 \times 10^6$ |
| 6.0 | 13.3% | 58.6% | $3.05 \times 10^6$ |
| 6.6 | 14.7% | 53.9% | $2.79 \times 10^6$ |
| 7.2 | 16.0% | 49.6% | $3.05 \times 10^6$ |
| 7.5 | 16.7% | 47.9% | $3.18 \times 10^6$ |
| 10 | 22.2% | 35.8% | $4.17 \times 10^6$ |
| 13 | 28.9% | 29.8% | $4.92 \times 10^6$ |
| 20 | 44.4% | 24.2% | $5.90 \times 10^6$ |
| 27 | 60.0% | 21.2% | $6.09 \times 10^6$ |
| 43 | 95.6% | 17.3% | $4.06 \times 10^6$ |

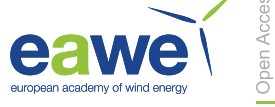


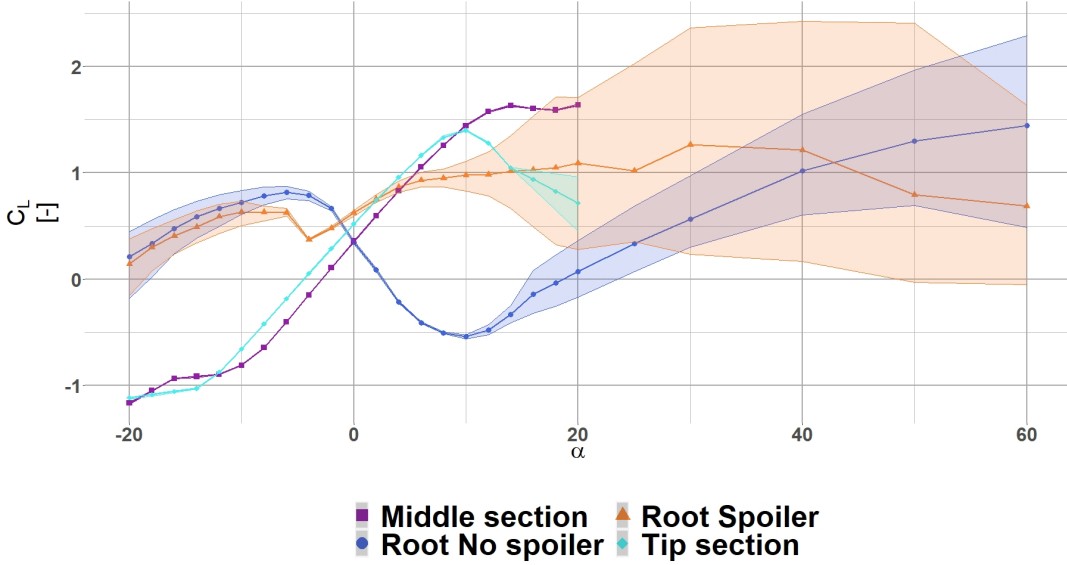

**Figure 6.** The blue dot (●) shows the $C_L$ for a representative root section without spoiler, the orange triangle (▲) shows the $C_L$ for a representative root section with spoiler, the purple square (■) shows the $C_L$ for a representative middle section and the cyan diamond (◆) shows the $C_L$ for a representative tip section. The plotted polars have not been corrected with a rotational model.

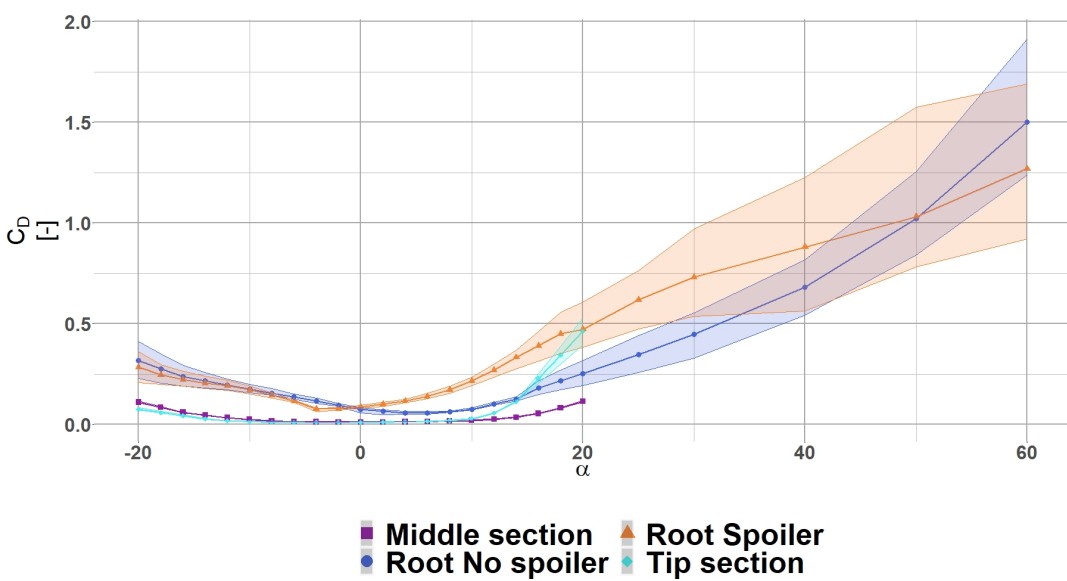

**Figure 7.** The blue dot (●) shows the $C_D$ for a representative root section without spoiler, the orange triangle (▲) shows the $C_D$ for a representative root section with spoiler, the purple square (■) shows the $C_D$ for a representative middle section and the cyan diamond (◆) shows the $C_D$ for a representative tip section. The plotted polars have not been corrected with a rotational model.





## 2.5 BEM simulations set-up

The following sections will detail the models set up used during the aeroelastic simulations. The first goal of the present paper is to determine the maximum aerodynamic potential of spoilers compared to a bare blade, free of any constraints. A second goal is to assess the impact of the spoiler, on the turbine lifetime, when running at maximum power extraction.

### 2.5.1 Pitch settings for maximal power production

The pitch settings for maximum power extraction are unknown, the turbine manufacturers may not recommend maximum power generation pitch settings due to potential noise, stall or load issues. Therefore, using SCADA measurements is not sufficient, an optimisation study is necessary. In order to reduce the optimisation space to only a single variable (the pitch settings), we assume that the turbine's rotational speed available thanks to averaged field measurements is optimised and will

not vary. Then, a search for the optimum pitch settings was carried out for each wind speed between cut-in (3m/s) and cut-out (20m/s), by increment of 0.5m/s, and for each turbine configuration (see Table 1). The optimisation constraints are described as: below rated wind speed (here 10.5m/s), the power production has to be maximal, whilst from rated wind speed until cut-out the turbine must regulate the generated power in order to maintain rated power (here 2.05MW). A sweep of pitch settings for a range between $-10°$ and $10°$ below rated and between $0°$ and $20°$ above rated was tried. The Figure 8 shows the outcome of

the trials for the turbine with spoiler using the mean aerodynamic coefficient polar: each curve represents a tested wind speed and pitch setting. The black stars shows the pitch settings for maximal power production and will be used during the rest of the study.

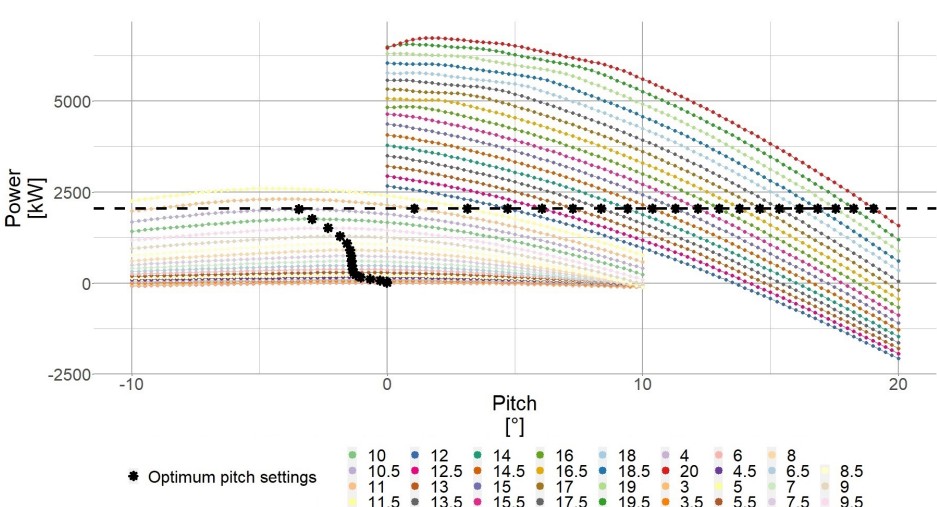

**Figure 8.** Power response with varying pitch settings for different wind speed. Each coloured dotted lines represents a wind speed. The black stars (∗) represent the pitch settings for maximum power production. The horizontal black dashed line is the turbine's rated power.



### 2.5.2 Rigid turbine simulations

In the first analysis, the turbine is considered rigid (i.e. not flexible) with the hub height 80m above ground using the standalone
AeroDyn module. The aerofoils associated to the CFD calculated polars are precisely defining the blade discretisation as
detailed in the Table 2. As the standalone AeroDyn module can only simulate steady wind profiles, we chose to use the power
law relation as seen in equation 3:

$$U(Z) = U \times \left( \frac{Z}{H_H} \right)^{\kappa} \tag{3}$$

Where, $U(Z)$ is the vertical wind speed, $U$ is the reference wind speed at a hub height, $Z$ is the height varying between the
ground and the top of the turbine, $H_H$ is the hub height and $\kappa$ is the wind shear exponent (here 0.2).
The air density in the BEM calculations is considered constant in space and time and is equal to the one used for the CFD polar
calculation ($\rho = 1.225$).

### 2.5.3 Flexible turbine simulations

The second analysis is a fully flexible turbine with turbulent wind using OpenFAST. The tool TurbSim (see Jonkman and Buhl
(2005)) developed by NREL is used to generate 10 minutes long three dimensional turbulent wind fields for each wind speed.
The box representing the wind field is is 150m wide and high subdivided in 50 points and 600s long. The IEC Kaimal Model
is used as spectral model thanks to the directly available IEC class requirements (here IEC class II chosen). The underlying
assumption is that the atmospheric conditions are considered neutral following the Monin-Obukhov similarity theory as detailed
in Wharton and Lundquist (2012); Holtslag et al. (2014-12-16); Diaz et al. (2010)

## 3   Results

After running all the turbine configurations a deep aerodynamic analysis is possible as many sensors outputs are available. For
brevity reasons only a small sample of all the available results will be presented. The multiple polar "states" (mean, maxi, mini)
allow to assess the variation around the mean value giving a measure of unsteadiness. First, the rigid turbine loads, power and
AEP are analysed. Secondly, the flexible turbine fatigue impact is analysed.





## 3.1 Rigid turbine


In the Figure 9 to Figure 11, the x-axis represents the blade radius, the y-axis represents the considered sensor output and each subplot represents a wind speed whose value is given in the title, from cut-in (3m/s) to rated power (10.5m/s).

### 3.1.1 Aerodynamic parameters

The lift coefficient of the "no spoiler" case is showing very low values inboard, as expected from very thick aerofoils. After the

radial position R7.2 ($\frac{r}{R} = 16\%$), both curves merge describing the end of the spoiler effect. For the "spoiler" case, the mean $C_L$ increases to relatively high values, especially for such inboard sections, ($C_{L_{mean}}^{spoiler} \approx 1$). However, the associated variation increases drastically. Indeed, the variation for the "no spoiler" case was $C_{L_{mean}}^{nospoiler} \pm 0.3$ while in the "spoiler" case the variation is close to $C_{L_{mean}}^{spoiler} \pm 1$ (see Figure 9). A similar outcome is seen for the drag coefficient (not shown here). The large variation in $C_L$ is a consequence of the polar unsteadiness.

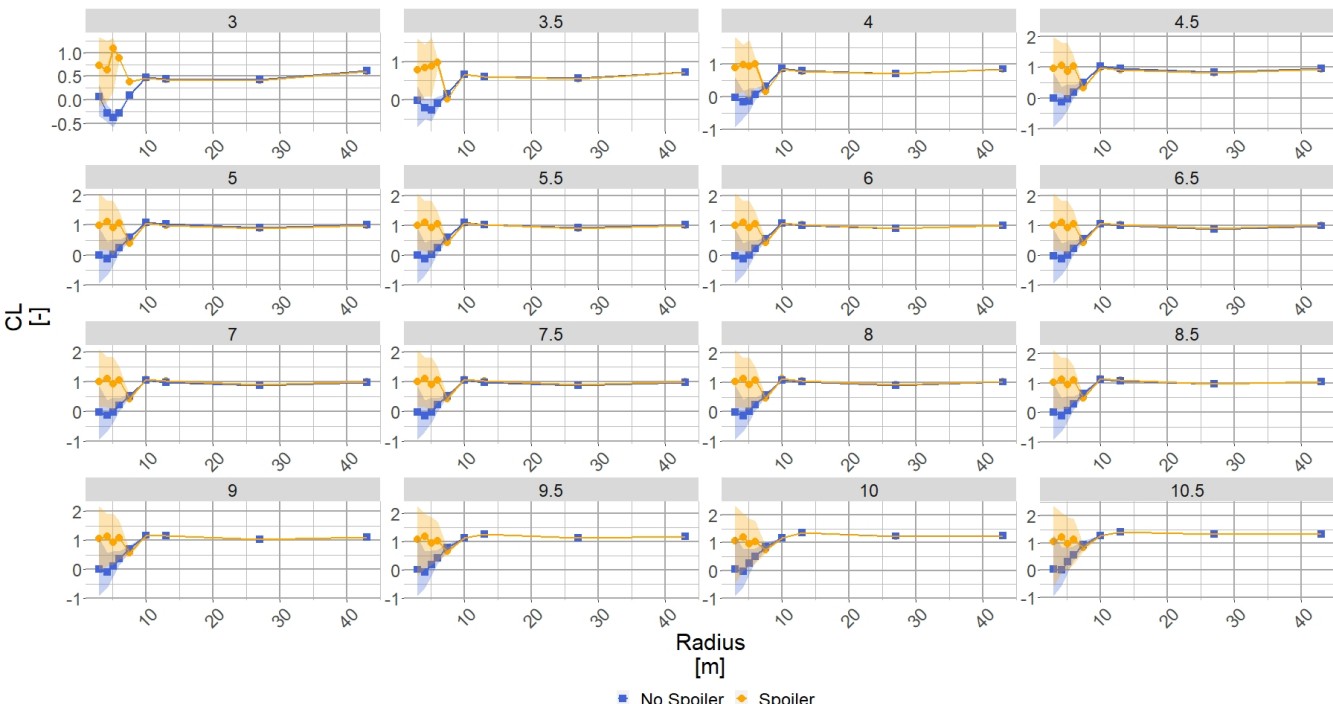

**Figure 9.** The blue square (■) shows the $C_L$ evolution along the blade radius without spoiler, the orange dot (●) shows the $C_L$ evolution along the blade radius with spoiler. Each subplot shows the results for a wind speed (m/s) whose value is given in the title.





The axial induction, $a$, is a key aerodynamic metric for turbine analysis. Through this parameter it is possible to have information about the sectional energy extraction and the sectional turbine loading. The energy extraction is at its maximum when $a = \frac{1}{3}$, according to the Betz's limit, and the loads increase significantly beyond $a = 0.4$ following the highly loaded rotor relationship (Glauert correction). Therefore, most turbine manufacturers aim for an induction factor value close to the optimal $a = \frac{1}{3}$, when in power production mode. After the pitch optimisation, the turbine runs close to optimal axial induction

for the outer part of the blade.

The "no spoiler" case show very low induction values at the root of the blade due to the cylinder shape: low lift coefficient and high drag values. The blade's inboard is not efficient to extract energy but the expected load level is consequently low. Where the spoiler is installed the induction increases and similarly to the lift coefficient the upper band of the variation due to the polar unsteadiness is close to the optimal induction. The average induction level at the spoiler location is close to $a = 0.2$, which is a

significant improvement compared to the "no spoiler" case where the induction level is close to 0 (see Figure 10). The relative variation area is similar compared to the lift coefficient: $a_{mean}^{spoiler} \pm 0.1$, and still a lot larger than the "no spoiler" case.

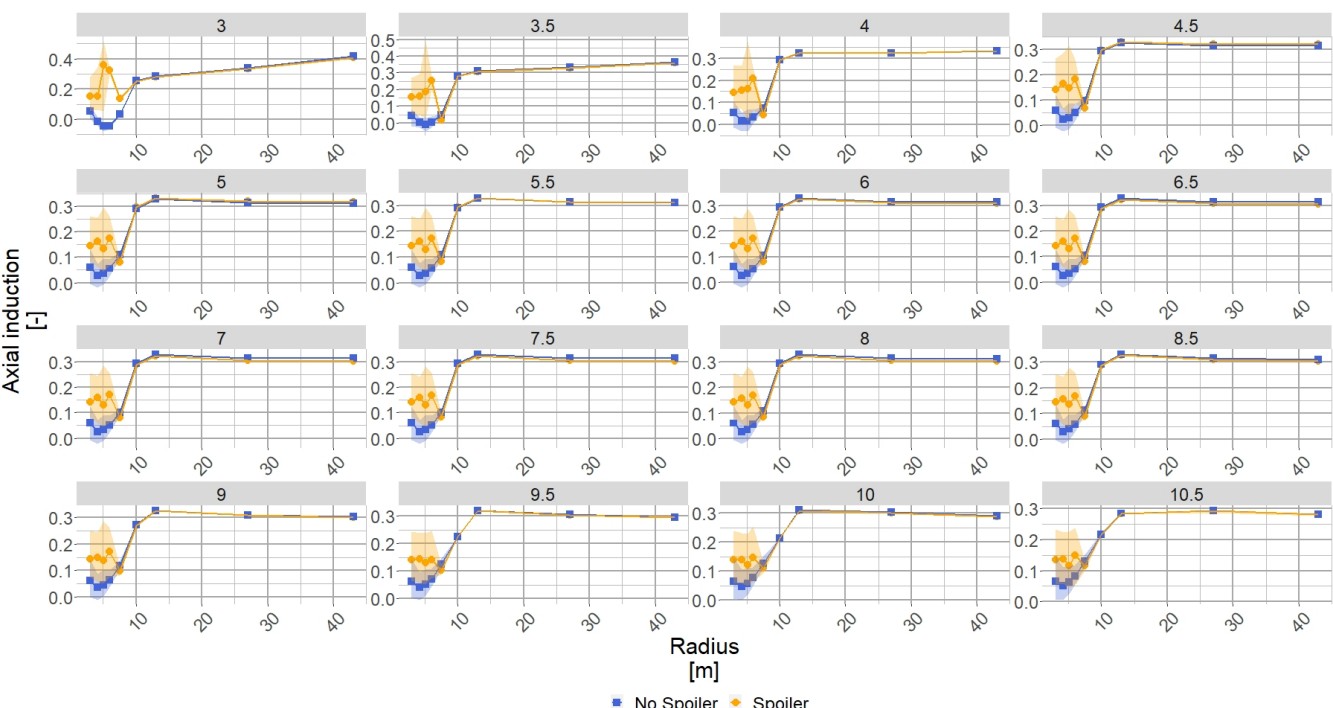

**Figure 10.** The blue square (■) shows the $a$ evolution along the blade radius without spoiler, the orange dot (●) shows the $a$ evolution along the blade radius with spoiler. Each subplot shows the results for a wind speed (m/s) whose value is given in the title.





### 3.1.2 Aerodynamic static loads

The local out-of-plane force ($F_X$) is calculated and its evolution against the radius for several wind speed is shown in Figure 11. Through the momentum theory and the axial induction, $F_X$ is directly proportional to the power production. The bare blade
design intent showed very low normal forces at blade root level with an almost constant increase along the span past R10. After the spoiler installation the local force increases significantly, despite being significantly lower than the outer part of the blade.

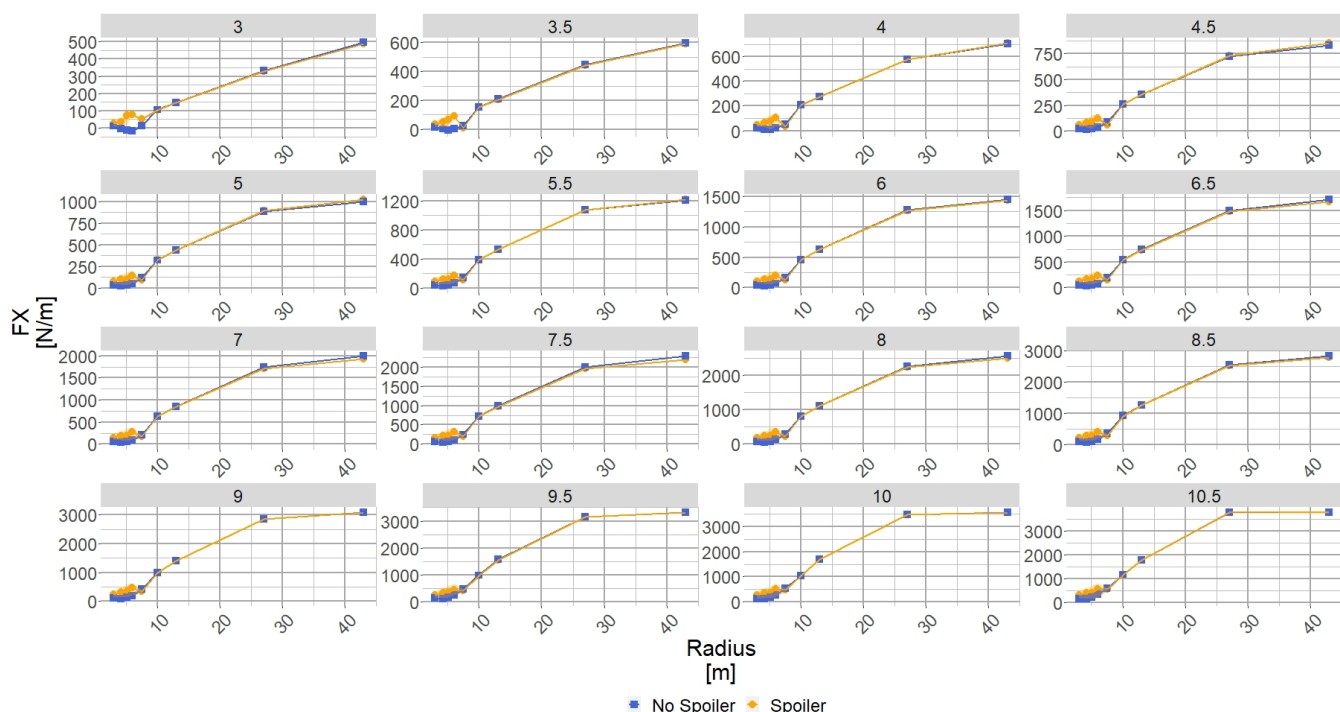

**Figure 11.** The blue square (■) shows the force normal to the rotor plane evolution along the blade radius without spoiler, the orange dot (●) shows the the normal force to the rotor plane evolution along the blade radius with spoiler. Each subplot shows the results for a wind speed ($\mathrm{m/s}$) whose value is given in the title.

The previous figures showed the results at aerofoils level, the next phase of the analysis will focus on the integrated values.





### 3.1.3 Integrated load: Root Bending Moment

The unsteadiness caused by the spoiler doesn't seem to be reflected at rotor level, the coloured area around the mean value
is almost nonexistent. Also, because the change is very small, both curves looks like they are overlapped in Figure 12. The
vertical bars show the difference between the mean RBM values for the "spoiler" and "no spoiler" case. Except around 5m/s,
the use of a spoiler tends to decrease slightly the RBM (right hand side vertical axis).

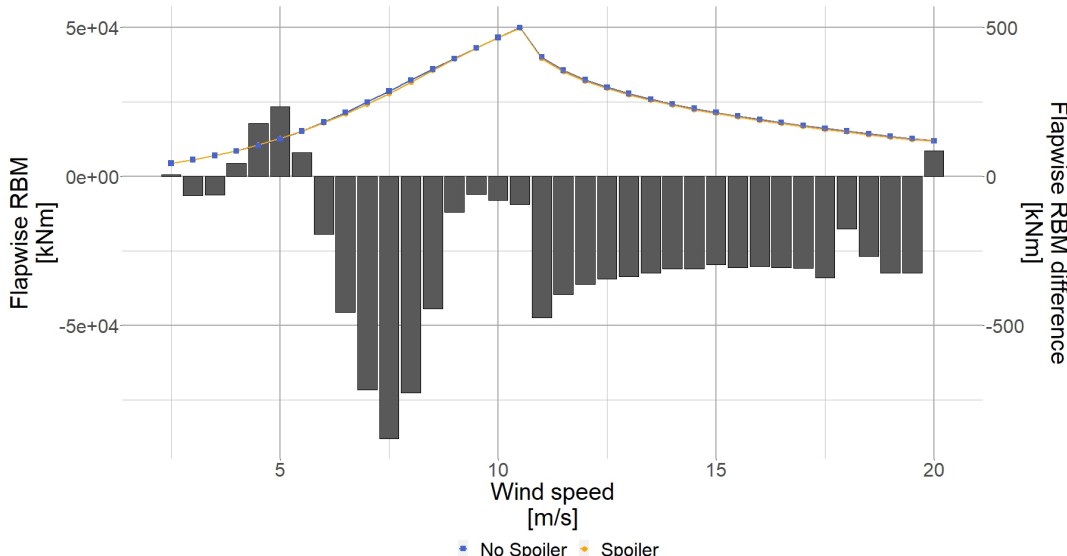

**Figure 12.** The blue square (■) shows the flapwise root bending moment evolution without spoiler, the orange dot (●) shows the flapwise
root bending moment evolution with spoiler. The black bars show the difference between the "spoiler" and "no spoiler" case for each wind
speed.





### 3.1.4 Power curve and energy production

The mean power curves for the "no spoiler" and "spoiler" configuration can be plotted (see Figure 13). Both curves are very
close to each other, the vertical bars shows that the "spoiler" does produce more energy on average, albeit a small amount
(power difference on the right hand axis). The error bars show the variation in power due to the different polar states used, i.e.
the top of the error bar is the power difference between the "spoiler" and the "no spoiler" case using the maximum aerodynamic
coefficients polar.

It is to be noted that, interestingly, the power gain of approximately 1% across the range of wind speeds is similar to the $C_L$
gain thanks to the spoiler presented in Figure 9.

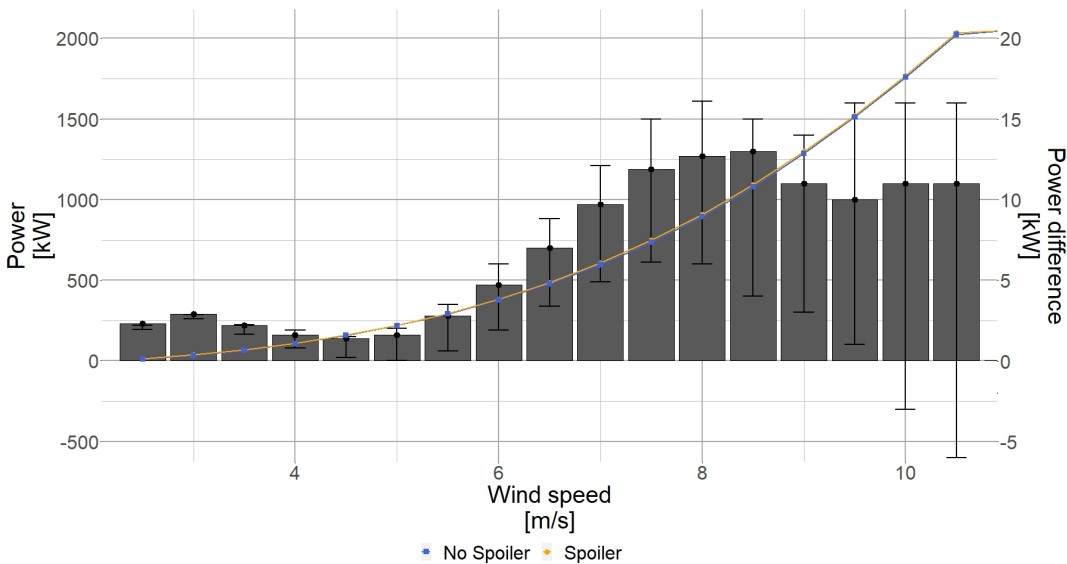

**Figure 13.** Power curve close-up for the low wind speeds. The blue square (■) shows the power curve without spoiler, the orange dot ●)
shows the power curve with spoiler. The black bars show the difference between the mean "spoiler" and mean "no spoiler" case.

After integrating the mean power curves over a year simulating a wind site condition IEC class II (Weibull shape factor =
0.2 and average wind speed = 8.5m/s), the AEP impact can be seen in the Table 3. On average, the spoiler produces 0.49%
AEP more than the "no spoiler" case, assuming maximum power extraction settings.

**Turbine unsteadiness definition**

When using BEM, one cannot use a time varying description of each angle of attack during the iterative procedure. Using
several steady states polars representing the different possible aerodynamic coefficients allowed for a first estimation of the
variation due to the unsteadiness. Analysing the loads or the different aerodynamic metrics (such as presented in Section 3.1.1
and Section 3.1.2) using three different polar states independently is acceptable because the data represents instantaneous
"snapshots" values. Also the steady wind profile used allows reproducible and repeatable results. However, to integrate the



**Table 3.** Spoiler impact on the AEP

| Turbine configuration | AEP [MWh] | AEP gain ratio [%] |
|---|---|---|
| No spoiler | 8256.5 | N/A |
| Spoiler | 8269.9 | 0.49 |

results in time, to calculate the mean thrust or the AEP, this assumption cannot hold. Indeed, assuming that the aerodynamic coefficients time variation is periodic, as illustrated in Figure 2, then after integration all oscillations cancel out. Therefore, the unsteadiness caused by the spoiler on time integrated quantities cannot be assessed. For this reason, the following method has been applied to give a measure of the variation caused by the spoiler, using the AEP as example.

The total variation of power for each wind speed is found by: $\Delta P_{WS} = P_{WS_{max}} - P_{WS_{min}}$. Then, knowing the Weibull site
characterisation it is possible to calculate the probability of each wind speed occurring over a year: $pr(WS)$. Combining both information the weighted Weibull average total variation around the mean value is found (see Equation 4).

$$\delta P = \sum_{WS=3}^{WS=20} \Delta P_{WS} \times pr(WS) \tag{4}$$

Where $\delta P$ is the Weibull weighted average power variation, $\Delta P_{WS}$ is the power range over a wind speed, $WS$ is the considered wind speed, and $pr(WS)$ the wind speed occurrence probability.

**Table 4.** Spoiler total variation around the mean value

| Turbine configuration | AEP variation [MWh] | Thrust variation [kN] |
|---|---|---|
| No spoiler | 27.6 | $20.5 \times 10^3$ |
| Spoiler | 70.4 | $22.7 \times 10^3$ |

Table 4 shows that the spoiler addition increase the inherent variation around the mean value for the AEP and the average aerodynamic thrust.

### 3.2   Flexible turbine

As seen in the previous sections, the rigid modelling shows little AEP benefit of installing the spoiler. However, due to the large increase in the mean local loads and its associated variation introduced by the spoiler it seems interesting to investigate the
damage and fatigue on the turbine. The aeroelastic calculations will be performed by OpenFAST with a fully flexible turbine. The fatigue analysis will focus on the blade only but can be extended to the whole turbine.



### 3.2.1 Combination method

A method to account for unsteadiness on a rigid turbine has been presented in Section 3.1.4, it can only simulate integrated
load. In order to analyse further the spoiler unsteadiness impact fatigue analysis is necessary. However, the same BEM limi-
tations arise. Here again, we chose to calculate each configuration ("no spoiler" and "spoiler") using the three polars for each
wind speed (from 3m/s to 20m/s). Then, thanks to a previously calculated Vortex Shedding Frequency (VSF) for each aerofoil
section a new time series is generated, as detailed below.

**Vortex Shedding Frequency (VSF)**

In Potentier et al. (2021b), the authors showed that a VSF can easily be found for a single aerofoil at a single angle of attack
using 2D CFD velocity (or load) time series. Applying the same methodology for all aerofoils and for all angles of attack, 2D
CFD load time series were post processed, thereby creating a database of VSF (see Figure 14). Using the BEM hypothesis of
2D flow, we assume that neither the blade rotation nor the blade deflection change the VSF. Moreover, the turbulent wind speed
frequency spectrum is independent from the VSF, we can therefore perform the interpolation in the time domain between time
series rather than in the frequency domain.

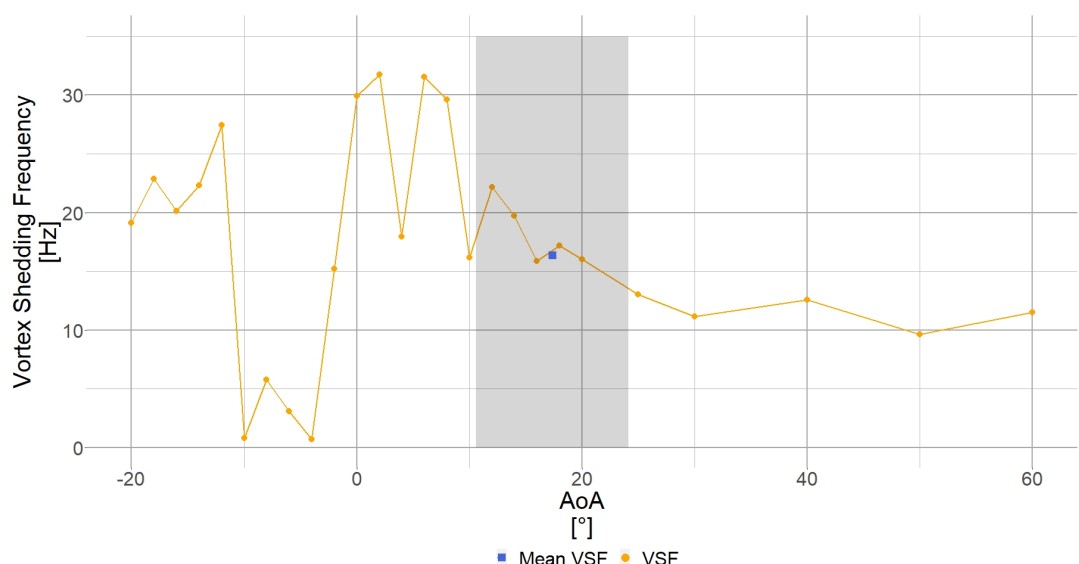

**Figure 14.** Plot of the angle of attack versus the Vortex Shedding Frequency for the "spoiler" case radial position R6 ($\frac{r}{R} = 13.3\%$) calculated
in 2D CFD. The blue square (■) shows the mean VSF (interpolated VSF at the mean angle of attack of the time series), the orange dots (●)
show evolution of the VSF depending on the angle of attack. The shaded area shows the standard deviation of the angle of attack time series.





Because of the sampling theorem, the OpenFAST sampling output rate must be at least twice higher than the highest VSF. The highest calculated VSF of all sections is approximately 60Hz. To add safety margin, the OpenFAST output is set to be at 160Hz, which is equivalent to a time step of $\Delta t_{OF} = 0.0063$s.

**New OpenFAST time series creation**


Once all aeroelastic results are available (Figure 15a), a mean VSF ($VSF_{mean}$) is determined by interpolating at the mean angle of attack of the mean time series result the VSF (blue square on Figure 14). Inverting it, leads to a representative time step for the considered wind speed $\Delta t = \frac{1}{VSF_{mean}}$. Then, the original results calculated using max, mean or mini polar are interpolated at new time steps using $\Delta t$ (Figure 15b).


An intermediate time series is generated, for each sensor. Again, supposing a periodic variation of the lift and drag coefficients, we assume that the first aerodynamic coefficient "seen" by the aerofoil is from the maximum polar, it then changes to the mean polar and finally the minimum polar and varies following this cycle for 600s. Such behaviour leads to the creation of the pink curve on Figure 15c.

One final numerical manipulation is necessary because all intermediate time series created possess different VSF and there-


fore different $\Delta t$. By re-sampling them at the same OpenFAST sampling rate ($\Delta t_{OF} = 0.0063$s). Because the turbulent wind speed frequency spectrum is independent from the VSF, we can perform the interpolation in the time domain rather than in the frequency domain, we thus ensure possible further analysis (Figure 15d).

This method is repeated for each radial position, each wind speed and for all local loads. The results presented in the next sections are using the data generated by this method.

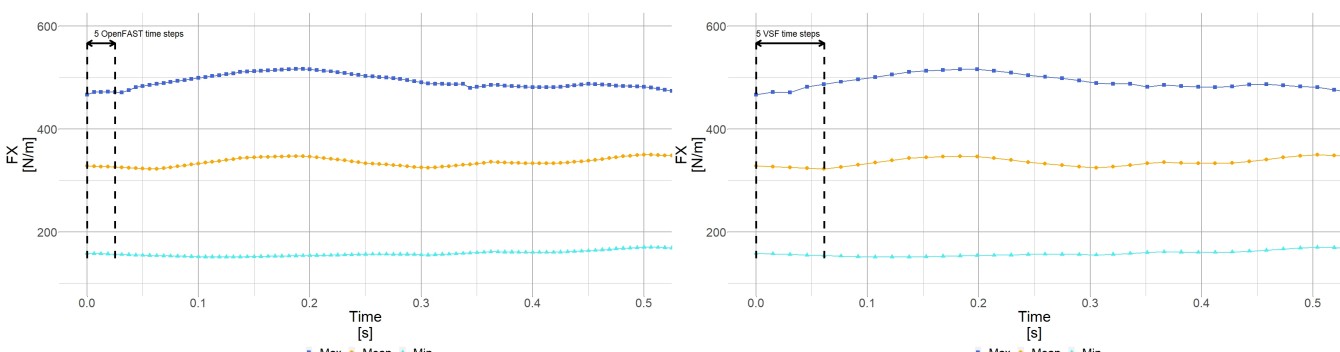

(a) OpenFAST time evolution of 0.5s of the local out-of-plane force ($F_X$) using the maximal, mean and minimal aerodynamic polar

(b) Interpolated OpenFAST results of the local out of plane force ($F_X$)

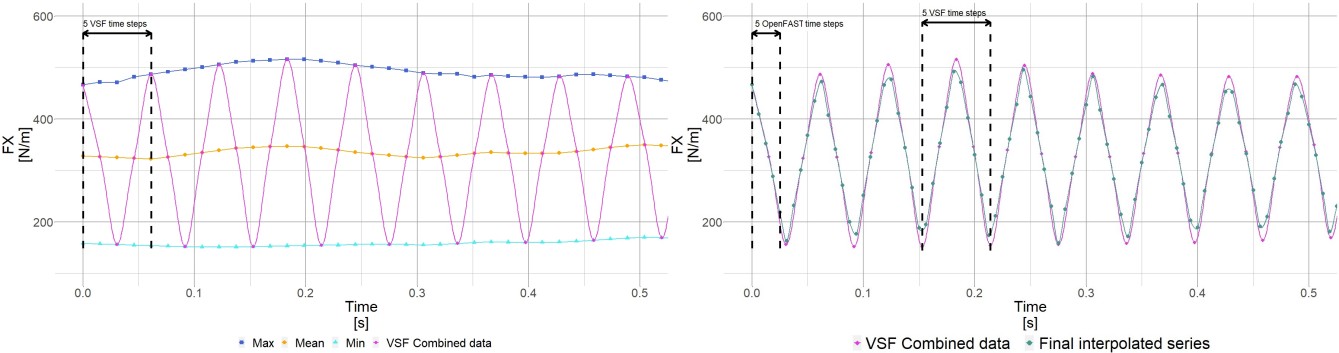

(c) Creation of the intermediate time series by alternating between the different interpolated time series.

(d) Generation of the final time series using the sampling rate from OpenFAST ($\Delta t_{OF} = 0.0063$s)

**Figure 15.** Illustration of the combination method using the local out-of-plane force ($F_X$) for 0.5s.



### 3.2.2 Normal force results

The Figure 16 shows the force normal to the rotor plane ($F_X$) for a 600s long OpenFAST simulation with an average horizontal wind speed of 8m/s (hub height). Each subplot shows a radial location, from R = 3.6 to R = 7.5 (from $\frac{r}{R} = 8\%$ to $\frac{r}{R} = 16.7\%$), the horizontal axis shows the time spent in the simulation. $F_X$ is clearly higher in the "spoiler" case regardless of the spanwise location.

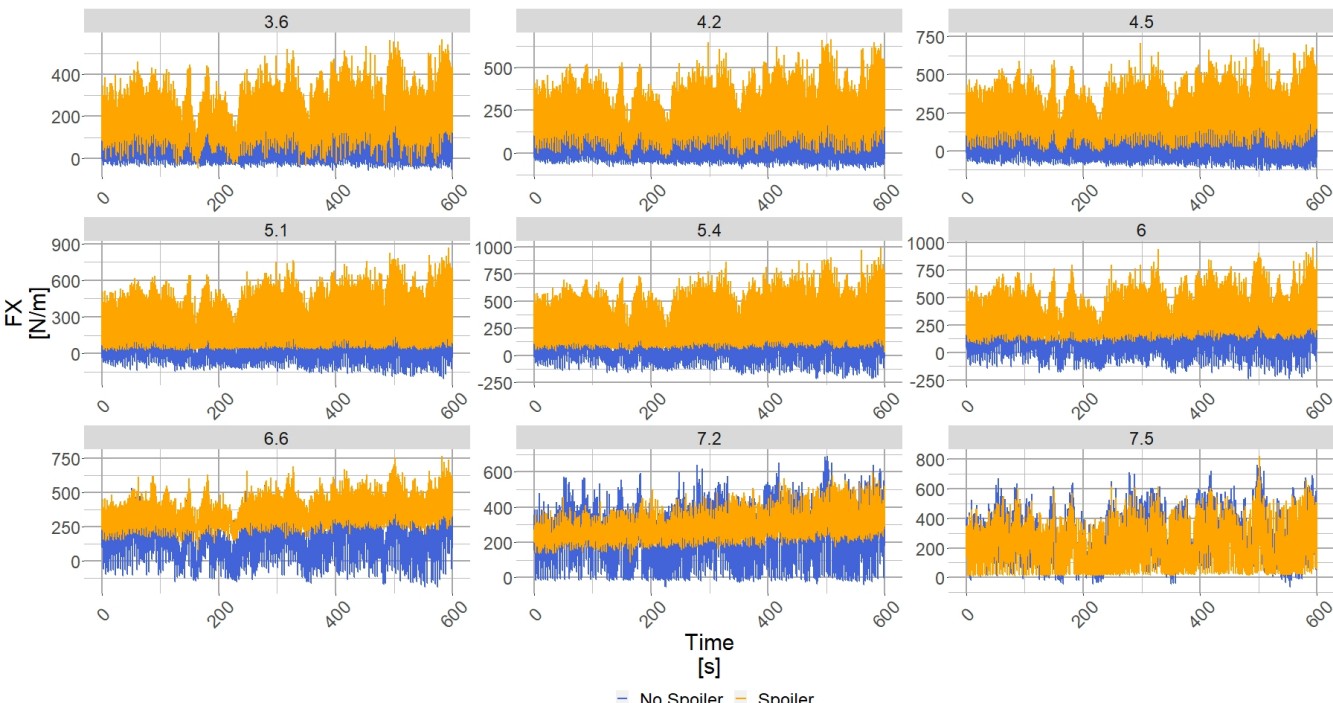

**Figure 16.** OpenFAST output normal force to the rotor plane for an average horizontal wind speed of 8m/s (hub height). The blue square (■) shows the blade results without spoiler using the combination method, the orange dot (●) shows the blade results with spoiler using the combination method. Each subplot shows the results for a radial location (m) whose value is given in the title.

The Figure 17 compares the Power Spectral Density (PSD) for the "spoiler" case using either the mean aerodynamic polar results or the combination method results. At low frequencies the PSD are overlapped, since the same turbulent wind speed was used in all aeroelastic simulations, however after the $VSF_{mean}$ is reached, the combination method shows clear peaks and harmonics. The curve trend is also showing the same downward behaviour at higher frequencies. The higher energy in the spectrum for the combination method hints at a higher fatigue loads for the combination method than using directly the OpenFAST results.

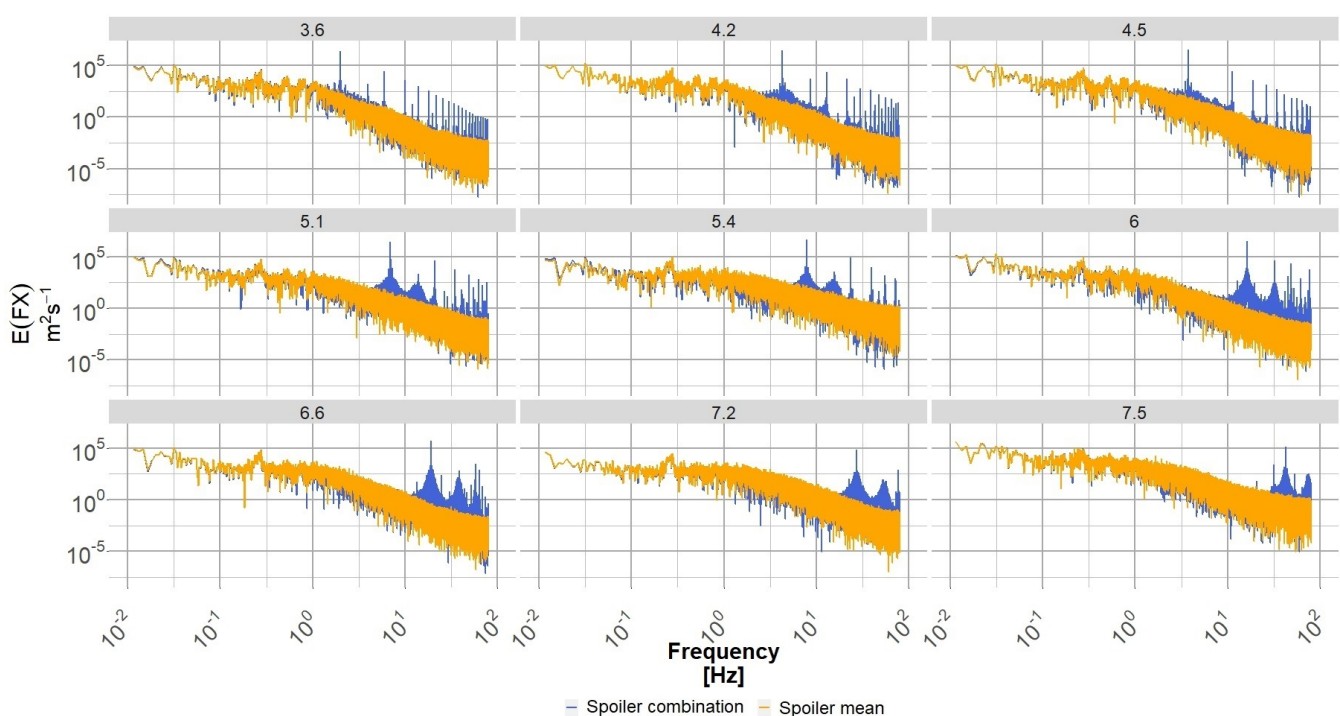

**Figure 17.** OpenFAST output normal Power spectral Density of the normal force to the rotor plane for an average horizontal wind speed of 8m/s (hub height). The blue square (■) shows the blade results with combination method for the spoiler, the orange dot (●) shows the blade results for the mean spoiler. Each subplot shows the results for a radial location (m) whose value is given in the title.



### 3.2.3 Fatigue results

After running in OpenFAST all wind speeds for both turbine configurations and generating the new time series as described in Section 3.2.1, it is possible to determine the lifetime impact of the spoiler and its associated unsteadiness on the turbine. The tool used is Mlife, also developed by NREL (see Hayman (2012)). Similarly to the AEP calculation, we assume that

the generated OpenFAST outputs follow a Weibull distribution of an IEC site B (shape factor = 2 and average wind speed = 8.5m/s).

In order to calculate the blade lifetime with the predefined pitch and RPM settings, an ultimate load before rupture for each analysed sensor must be given. Since the material properties are unknown, we used MExtreme (see Hayman (2015)) to extract the highest sectional loads of interest (here $F_X$ and $F_Y$) seen by the turbine of both cases as first approximation. To assess

the evolution of lifetime with respect to the ultimate load, three distinct load values were selected. Using those values for the "no spoiler" and "spoiler" case, it is now possible to plot the lifetime evolution with respect to the ultimate load for the local out-of-plane force (see Figure 18a) and the local in-plane forces (see Figure 18b). The horizontal dashed line shows the usual 25 years design life time. The lowest symbol of each coloured line represents the turbine lifetime if it was designed based on the highest load found by MExtreme. The following two points are calculated lifetimes using the initial highest load multiplied

by a factor 2 and 3 As expected, the behaviour is highly non-linear and can reach unrealistic lifetime expectancy. To avoid running fatigue simulation with a very low life expectancy, we choose the loads from the multiplication factor of 2 as baseline for the rest of the analysis (see Appendix B). The Wöhler exponent was kept constant throughout the study to a representative wind turbine blade material: m = 10 (see Lloyd (2010)).

Because of the different hypotheses taken, we are only analysing trends and not presenting the direct Mlife results. There-

fore a life index ($L_i$) is created by normalising the outputs of the "no spoiler" case to create a baseline, i.e. for each sensor $L_i^{nospoiler} = 1$. Then the "spoiler" case results are normalised by the previously created baseline. The Table 5 summarises the outcome of the fatigue calculation. The second column shows the life index impact of the considered sensor when adding a spoiler.

$$L_{ij}{}^{nospoiler} = \frac{T_{life_j}^{nospoiler}}{T_{life_j}^{nospoiler}} \qquad L_{ij}{}^{spoiler} = \frac{T_{life_j}^{spoiler}}{T_{life_j}^{nospoiler}} \tag{5}$$

Where $L_i$ is the life index of the "no spoiler" or "spoiler" case, $j$ is the section considered.

As suspected in the Section 3.1.2, all local forces see a negative impact after installing the spoilers, $L_i^{spoiler} < L_i^{nospoiler}$ indicates that this specific section will fail before the "no spoiler" turbine. Despite the hypotheses and assumptions, the method employed captures well the negative impact of the spoiler on the local sections, which is in line with the blade failures (cracks), seen at the spoiler's end, in the field by ENGIE Green maintenance team. It is to be noted that BEM aeroelastic simulations can

model, neither the spoiler's glue joint, nor the internal elements of the blade (such as spar or web). A dedicated Finite Element Analysis (FEA) would be required to answer the question fully, but such study is out of the scope of the present paper.

(a) Out of plane local load

(b) In plane local loads

**Figure 18.** Life time expectancy evolution with respect to the ultimate load chosen. The blue square (■) shows the blade results without spoiler, the orange dot (●) shows the blade results with spoiler and each symbol represent a blade nodal output ($8.0\% < \frac{r}{R} < 16.7\%$).



**Table 5.** Life index of the "spoiler" case

| Sensor output | Life index [-] | Description |
|:---:|:---:|:---:|
| B1N1Fx | 0.00 | r/R = 8.0% normal force (flap) |
| B1N2Fx | 0.00 | r/R = 9.3% normal force (flap) |
| B1N3Fx | 0.01 | r/R = 10.0% normal force (flap) |
| B1N4Fx | 0.01 | r/R = 11.3% normal force (flap) |
| B1N5Fx | 0.00 | r/R = 12.0% normal force (flap) |
| B1N6Fx | 0.01 | r/R = 13.3% normal force (flap) |
| B1N7Fx | 0.02 | r/R = 14.7% normal force (flap) |
| B1N8Fx | 0.03 | r/R = 16.0% normal force (flap) |
| B1N9Fx | 0.02 | r/R = 16.7% normal force (flap) |
| B1N1Fy | 0.01 | r/R = 8.0% tangential force (edge) |
| B1N2Fy | 0.01 | r/R = 9.3% tangential force (edge) |
| B1N3Fy | 0.01 | r/R = 10.0% tangential force (edge) |
| B1N4Fy | 0.00 | r/R = 11.3% tangential force (edge) |
| B1N5Fy | 0.00 | r/R = 12.0% tangential force (edge) |
| B1N6Fy | 0.00 | r/R = 13.3% tangential force (edge) |
| B1N7Fy | 0.11 | r/R = 14.7% tangential force (edge) |
| B1N8Fy | 0.30 | r/R = 16.0% tangential force (edge) |
| B1N9Fy | 0.59 | r/R = 16.7% tangential force (edge) |



To compare the results of the proposed method, the Table 6 shows the same life index calculation when using the steady polar. In some case the fatigue calculation predicts much higher residual lifetime when adding a spoiler. It appears in contradiction with the analysis performed so far. It highlights the risk of installing such AAO without knowing the aerodynamic impact and structural consequences.

**Table 6.** Life index of the "spoiler" case assuming steady polars

| Sensor output | Life index min polar [-] | Life index mean polar [-] | Life index max polar [-] | Description |
|---|---|---|---|---|
| B1N1Fx | 7.23 | 0.00 | 0.00 | r/R = 8.0% normal force (flap) |
| B1N2Fx | 4.16 | 0.00 | 0.00 | r/R = 9.3% normal force (flap) |
| B1N3Fx | 7.79 | 0.12 | 0.00 | r/R = 10.0% normal force (flap) |
| B1N4Fx | 3.07 | 1.79 | 0.00 | r/R = 11.3% normal force (flap) |
| B1N5Fx | 0.08 | 0.27 | 0.00 | r/R = 12.0% normal force (flap) |
| B1N6Fx | 0.16 | 0.41 | 0.00 | r/R = 13.3% normal force (flap) |
| B1N7Fx | 0.19 | 18.74 | 0.83 | r/R = 14.7% normal force (flap) |
| B1N8Fx | 0.37 | 704.85 | 1704.08 | r/R = 16.0% normal force (flap) |
| B1N9Fx | 0.70 | 2.73 | 2.05 | r/R = 16.7% normal force (flap) |
| B1N1Fy | 0.00 | 0.01 | 0.00 | r/R = 8.0% tangential force (edge) |
| B1N2Fy | 0.00 | 0.01 | 0.00 | r/R = 9.3% tangential force (edge) |
| B1N3Fy | 0.00 | 0.02 | 0.00 | r/R = 10.0% tangential force (edge) |
| B1N4Fy | 0.00 | 0.31 | 0.01 | r/R = 11.3% tangential force (edge) |
| B1N5Fy | 0.00 | 0.20 | 0.01 | r/R = 12.0% tangential force (edge) |
| B1N6Fy | 0.00 | 0.42 | 0.02 | r/R = 13.3% tangential force (edge) |
| B1N7Fy | 0.01 | 3.22 | 1.02 | r/R = 14.7% tangential force (edge) |
| B1N8Fy | 0.00 | 287.13 | 372.88 | r/R = 16.0% tangential force (edge) |
| B1N9Fy | 0.01 | 3.42 | 2.58 | r/R = 16.7% tangential force (edge) |



## 4 Conclusions

The authors built an aeroelastic BEM model for a commercial turbine retrofitted with root spoilers using a 3D blade scan. It appears that the spoiler impact is minimal, both for the AEP or the rotor integrated loads. The AEP increases by a small margin ($\approx 0.5\%$) with a large variation associated while the integrated loads can increase by $\approx 1\%$. However, the local loads increase
significantly with a large variation around the mean value.

A fatigue analysis has been performed using a novel way of capturing the local unsteadiness due to the aerofoil's behaviour. It uses 2D CFD flow characteristics (Vortex Shedding Frequency) as well as the results calculated from three different steady polars (maximum, mean and minimum aerodynamic coefficients). The spoiler increases the already locally present unsteadiness and should not be neglected in the turbine's structural design. The spoiler can be detrimental to the turbine lifetime, retrofitting
such devices should be done with care and the mechanical properties should be re-evaluated prior to installing the spoiler.





*Code and data availability.* Available on demand.

## Appendix A: ENGIE Green turbine's characteristics

**Table A1.** Blade characteristics

| Metric | Value | Unit |
|---|---|---|
| Mass | 8100 | kg |
| Length | 45.2 | m |
| Maximum chord | 5 | m |
| Rotor diameter | 92.5 | m |

**Table A2.** Hub characteristics

| Metric | Value | Unit |
|---|---|---|
| Mass | 18700 | kg |
| Diameter | 4.5 | m |
| Height | 3.4 | m |
| Overhang | 1.89 | m |
| Mass moment of inertia about rotor axis* | 47334 | kg.m$^2$ |

*\* calculated*

**Table A3.** Nacelle characteristics

| Metric | Value | Unit |
|---|---|---|
| Mass | 69200 | kg |
| Length | 10.3 | m |
| Depth | 3.8 | m |
| Height | 3.9 | m |
| Mass moment of inertia about yaw axis* | 170982 | kg.m$^2$ |

*\* calculated*



**Table A4.** Drive train characteristics

| Metric | Value | Unit |
|---|---|---|
| Mass | 25646 | kg |
| Length | 4.9 | m |
| Depth | 3.0 | m |
| Height | 2.4 | m |
| Mass moment of inertia about high speed shaft axis* | 170982 | kg.m$^2$ |
| Gearbox ratio | 120 | - |

*calculated*

**Table A5.** Tower characteristics

| Metric | Value | Unit |
|---|---|---|
| Mass | 129700 | kg |
| Height | 80 | m |



## Appendix B: Ultimate loads

**Table B1.** Ultimate loads for various sensors

| Sensor output | Ultimate load | Description |
|:---:|:---:|:---:|
| B1N1Fx | 4124 | r/R = 8.0% normal force (flap) |
| B1N2Fx | 5002 | r/R = 9.3% normal force (flap)) |
| B1N3Fx | 4958 | r/R = 10.0% normal force (flap)) |
| B1N4Fx | 5407 | r/R = 11.3% normal force (flap)) |
| B1N5Fx | 6178 | r/R = 12.0% normal force (flap) |
| B1N6Fx | 6079 | r/R = 13.3% normal force (flap) |
| B1N7Fx | 4618 | r/R = 14.7% normal force (flap) |
| B1N8Fx | 4226 | r/R = 16.0% normal force (flap) |
| B1N9Fx | 4708 | r/R = 16.7% normal force (flap) |
| B1N1Fy | 3757 | r/R = 8.0% tangential force (flap) |
| B1N2Fy | 4067 | r/R = 9.3% tangential force (flap)) |
| B1N3Fy | 4190 | r/R = 10.0%tangential force (flap)) |
| B1N4Fy | 4595 | r/R = 11.3%tangential force (flap)) |
| B1N5Fy | 4924 | r/R = 12.0%tangential force (flap) |
| B1N6Fy | 4908 | r/R = 13.3%tangential force (flap) |
| B1N7Fy | 3255 | r/R = 14.7%tangential force (flap) |
| B1N8Fy | 2979 | r/R = 16.0% tangential force (flap) |
| B1N9Fy | 3665 | r/R = 16.7% tangential force (flap) |





*Author contributions.* TP performed the scans post-processing, CFD pre-processing and post-processing, BEM model building, calculations
and analysis and writing of the paper. EG performed CFD verification and helped set-up the CFD model. ClB and AF provided feedback
from the industrial point of view and CB helped with the proofreading of the manuscript and physical analysis of the results.

*Competing interests.* The authors declare that they have no conflict of interest.

*Acknowledgements.* The authors would like to acknowledge the ANRT (Association Nationale de Recherche Technologique) for their finan-
cial support.



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
