# Peer review of "High Reynolds number wind turbine blade equipped with root spoilers. Part II: Impact on energy production and turbine lifetime"

_Wind Energy Science, 2022_

## Referee Comment (RC1)

Page 4: The procedure to account for unsteady polars in OpenFAST is not clear. Did the authors simulate separate cases for high, medium and low lift? How would this allow one to account for the unsteadiness in the flow in BEM simulations? Also, is unsteadiness in Cd considered?

L95-100: Authors used the chord to scale stiffness properties of the NREL 5MW as opposed to thickness. Since the blades are made of different airfoils, perhaps thickess could be considered for flapwise and chord for edgewise scaling? In any case, blade stiffness properties seem to me as indicative due to the scaling procedure. In my opinion this odes not impact the validity of the study, but perhaps authors could consider stating this more clearly.

Figures 6,7 & following: Shaded areas are represenred in these figures. It is not clear from the legend what these areas refer to. Are they the upper and lower ranges of Cl and Cd? Same consideration for Figure 9: did the author run different sets of simulations in OpenFAS with high, low and medium polar coefficients and then shad the areas accordingly?

P11: Could the authors please better explain why a pitch optimization was necessary as opposed to using SCADA data? SCADA data will report how the manufacturer intended the blade to operate. Therefore, if the spoilers are intended as a retrofit, they should be evaluated with respect to the operating "baseline" blade. Also, is the optimization performed for the blade with or without root spoilers? Finally, a comment on the pitch values resulting from the optimization would be nice, since they are somewhat hard to grasp from the figure.

Figure 8: number of markers on the x axis can be increased to improve readability

Figures 10 & 11: are tip losses accounted for? I would expect axial induction to go up at blade tip due to Pradtl's tip loss correction and axial force to drop off.

Section 3.1.3: the decrease in root bending moment despite the increases in lift at root is interesting. How would the authors explain this? In a controller used in these simulations? Are there slight variations in rotor speed wich would cause slight differences in forces in the outer parts of the blade (not appreciable in figure 11)? In other words, figure 11 seems to show an increase in axial force, while in figure 12 a decrease in rotor therus is predicted, how can this be explained?

L210: "It is to be noted that, interestingly, the power gain of approximately 1% across the range of wind speeds is similar to the CL gain thanks to the spoiler presented in Figure 9." However looking at figre 13 this constant 1% seems reasonable only up to 8m/s

Table 4: It would be interesting to present values also as percentage respect to mean

Section 3.2.3: In the reviewers opinion, fatigue results should be investigated more in depth. For instance it would be interesting to evaluate the impact on root bending moment and not only on the sectional stresses at root.

---

## Author Response (AR1)

**Reviewer 1**

Authors analyze the effects of root spoilers on wind turbine performance and fatigue lifetime. A novel methodology to account for the unsteadiness introduced by the spoilers in Blade Element Momentum calculations is presented. The methodology appears to be reasonable, although it would be interesting to validate it with comparison to high fidelity models such as CFD. Authors should highlight this aspect clearly in the revised manuscript.

We thank the reviewer for the time spent reviewing the article and the feedback given.

A 3D CFD set-up of the study is being ran at the moment to verify the assumptions made regarding vortex shedding frequency.

The responses are written under each of the comments. For the responses below, the figure and table numbers are those from the revised manuscript.

**Comments**

1. Page 4: The procedure to account for unsteady polars in OpenFAST is not clear. Did the authors simulate separate cases for high, medium and low lift? How would this allow one to account for the unsteadiness in the flow in BEM simulations? Also, is unsteadiness in Cd considered?
Yes, we ran 3 cases (maximum, mean and minimum) for each turbine configurations (spoiler and no spoiler) in OpenFAST, as per the Table 1. The unsteadiness is accounted for in the results post-process as presented in Section 3.1 for the rigid turbine and in Section 3.2 for the flexible turbine. The unsteadiness is highlighted in the rigid turbine results by the shaded area around the solid line (mean value). For the flexible turbine, the unsteadiness is accounted for in the fatigue calculation presented in Section 3.2.3. The Cl, Cd and Cm unsteadiness are considered.

2. L95-100: Authors used the chord to scale stiffness properties of the NREL 5MW as opposed to thickness. Since the blades are made of different airfoils, perhaps thickess could be considered for flapwise and chord for edgewise scaling? In any case, blade stiffness properties seem to me as indicative due to the scaling procedure. In my opinion this odes not impact the validity of the study, but perhaps authors could consider stating this more clearly.
It is a valid point and we didn't think of the possibility of "splitting" the scaling based on the direction. We will add the following in the manuscript:
"Also, the edgewise stiffness could have been scaled based on the chord thickness and thickness for the flapwise stiffness. It was decided to only use the chord as basis for the scaling for simplicity. Further studies could be done to assess the validity of the assumption."

3. Figures 6,7 & following: Shaded areas are represenred in these figures. It is not clear from the legend what these areas refer to. Are they the upper and lower ranges of Cl and Cd? Same consideration for Figure 9: did the author run different sets of simulations in OpenFAS with high, low and medium polar coefficients and then shad the areas accordingly?
It is correct, the legend in the Figure 7 and 8 is incomplete. The shaded area represents the maximum and minimum aerodynamic coefficients reached for each angle of attack. We will add it for the next revision. It is however mentioned in L123 to L126.
Regarding the Figure 11 and following, the solid line has been calculated using the "mean polar" while the shaded area was calculated with the "maximum" and "minimum polar".

4. P11: Could the authors please better explain why a pitch optimization was necessary as opposed to using SCADA data? SCADA data will report how the manufacturer intended the blade to operate. Therefore, if the spoilers are intended as a retrofit, they should be evaluated with respect to the operating "baseline" blade. Also, is the optimization performed for the blade with or without root spoilers? Finally, a comment on the pitch values resulting from the optimization would be nice, since they are somewhat hard to grasp from the figure.

The SCADA data is not considered reliable in this case for two main reasons: the pitch is read and compared against 10min average wind speed, the error bar associated to the mean value is large despite using yearly data. The second reason is that the pitch read is relative to the blade position which may (or not) be perfectly positioned. We did try using the pitch from the SCADA data, but after rated power it did not reach its nominal power. Moreover, the aim of the study is to find the absolute maximum gain of spoilers without any constraints, the turbine manufacturers' impose some constraints to avoid: loads, stall, noise, mechanical issues, etc. In our case we wanted to calculate the theoretical maximum power production free of constraints. The optimisation has been performed for both cases: with and without spoiler. The difference between both blade is very small. We will add to the manuscript a table detailing the pitch data with respect to the wind speed.

5. Figure 8: number of markers on the x axis can be increased to improve readability

The Figure has been replaced by the response surface from the optimisation procedure and the optimal pitch settings for all configurations.

6. Figures 10 & 11: are tip losses accounted for? I would expect axial induction to go up at blade tip due to Pradtl's tip loss correction and axial force to drop off.

Yes, the tip losses module was switched ON during the AeroDyn simulation. The last data point presented is at R43, which is 2m before the blade tip. It explains why the axial induction is still level, the "de-twist" happens at approximately R45 on this blade.
Also the presented data show the aerodynamic parameter before rated power where the blade has not started pitching to limit the generated power.

7. Section 3.1.3: the decrease in root bending moment despite the increases in lift at root is interesting. How would the authors explain this? In a controller used in these simulations? Are there slight variations in rotor speed wich would cause slight differences in forces in the outer parts of the blade (not appreciable in figure 11)? In other words, figure 11 seems to show an increase in axial force, while in figure 12 a decrease in rotor therus is predicted, how can this be explained?

There is indeed tiny differences in Rotation Per Minute (RPM) between the no spoiler and spoiler cases: approximately 0.1 RPM difference. We do not believe it explains the difference in RBM. However, the pitch settings being slightly different between the two configurations play a more important role. In the spoiler case, the pitch settings are less "aggressive" due to the higher power produced thanks to the blade inboard. Which means that for the no spoiler the pitch is towards higher CL values leading to a blade outboard more heavily loaded (which was not visible on Figure 11). Consequently, after integration, the RBM is more important for the no spoiler case. This is an important finding which will be highlighted in the next revision of the manuscript. The Figures from Section 3.1 have been remade to reduce the number of wind speeds shown. The explained behaviour is now clearly visible.

8. L210: "It is to be noted that, interestingly, the power gain of approximately 1% across the range of wind speeds is similar to the CL gain thanks to the spoiler presented in Figure 9." However looking at figre 13 this constant 1% seems reasonable only up to 8m/s

That is correct, the sentence will be modified as follow:

"It is to be noted that, interestingly, the power gain of approximately 1\% across the range of wind speeds is similar to the $C_L$ gain thanks to the spoiler presented in Figure 11, up to 8m/s. Closer to rated power, the power gain reduces."

9. Table 4: It would be interesting to present values also as percentage respect to mean
We added the percentage in the Table 5 and decided to remove the Thrust column in order to dive deeper into the AEP analysis.

10. Section 3.2.3: In the reviewers opinion, fatigue results should be investigated more in depth. For instance it would be interesting to evaluate the impact on root bending moment and not only on the sectional stresses at root.
So far, the method developed can only account for sectional loads since it relies on vortex shedding frequency. The RBM being an integrated load we cannot link it a particular frequency, since the vortex shedding frequency changes along the blade radius. For this reason we decided to focus only on sectional loads. The following has been added to the manuscript:
"The method developed can only account for sectional loads since it relies on vortex shedding frequency. The integrated load such as RBM, cannot be associated to any particular VSF."

**Reviewer 2**

The paper reports a computational study on the use of blades equipped with spoilers on the root section to increase the local lift coefficient. This is the second part of a unique work; in this second part the authors consider the energetic and structural implications of using spoiler blades. Authors conclude that the use of spoilers provides only a very marginal increase of wind energy harvesting, while reducing in a very significant way the life expectancy of the blades, due to the excess of unsteadiness caused by the augmented vortex shedding phenomena associated to the spoiler.

This conclusion is not very surprising, considering the aerodynamics of the profile with spoiler. This paper has the merit to quantify the impact of the spoiler in a multi-disciplinary way, and it is appreciated for its scientific quality. However, for the reader the motivation of the paper it is not clear: is the intention of the paper to demonstrate that the root-spoiler blades are not recommended, and should be avoided? Why did they decide to study this specific configuration, and how they designed it? Did they find something tunexpected, and why? At what extent these conclusions can be generalized? Without answer to these questions, the paper remains a documentation of a failed attempt of improving the design of wind turbine and, as is, it is of relatively low technical relevance (still having high scientific quality). This referee recommends to extended the motivation and conclusion sections of this paper to enhance its engineering impact.

We thank the reviewer for the time spent reviewing the article and the feedback given. Below are the answers to the questions asked.

The motivation behind the paper is to provide explanation to a real-life case which happened to ENGIE Green, a French exploiting party. After installing the spoiler, the maintenance team noticed that the blades were cracked. The cracks happened rapidly after the spoiler retrofit. The authors did not design the spoilers, we scanned it from a grounded blade. We did not aim at warning against root spoiler or any other add-ons, but answer a real-life issue faced by the exploiting party. The aim of the paper was to see if using simulation tools, it is possible to detect blade failures due to the unsteadiness caused by the spoilers. The Life index and associated fatigue calculation method is a first step in this direction.

For safety reasons, the outcome can be extended to any aerodynamic add-on or blade shape (such as flatback) producing this amount of unsteadiness. However, dedicated studies would be necessary to quantify the impact of the spoiler height and chordwise position on the unsteadiness caused, which is outside the scope of the present study.

The motivations and conclusions sections will be modified accordingly to increase the clarity on the authors' objectives.

The responses are written under each of the comments. For the responses below, the figure and table numbers are those from the revised manuscript.

**Comments**

Further suggestions for revision are given below:

1. At the beginning of Section 2, please properly introduce the turbine (which is recalled several time in the following of the paper)

ENGIE Green being the exploiting party of the wind farms, they would prefer to keep the turbine unnamed if possible. However, we can precise the rotor length of 92m and a total height of 150m. Moreover, technical data related to the turbine is available in the appendices.

2. Line 60, page 3: why do you need the BET for calculating the angle of attack from blade twist/pitch?

The angle of attack is calculated as $\alpha = \phi + \theta$ where $\phi$ is the inflow angle and $\theta$ the sum of twist and pitch. The inflow angle is the angle between the rotor plane and the relative velocity. In order to calculate the inflow angle, the BET is needed because the blade is discretised in independent elements. The Momentum theory is needed to calculate the induced velocities, thus producing the so called Blade Element Momentum (BEM) theory.

3. Section 2.4: please consider to show the airfoil shape, with and without spoiler

Thank you for the comment we added the aerofoil shapes

4. Line 155, pag. 12: please note that U(Z) it is not the vertical wind speed, but the wind speed distribution along the vertical direction

Thank you for the correction, it has been corrected as following:

"Where, U(Z) is the wind speed distribution along the vertical direction, U is the reference wind speed at a hub height, Z is the height varying between the ground and the top of the turbine, H_H is the hub height and $\kappa$ is the wind shear exponent (here 0.2)."

5. Figures 9-11: I recommend to show less plots, probably 4 or 6 are sufficient; in this way, their readability is highly improved

We reduced the number of plots as suggested.

6. Section 3.1.3: please introduce properly the topic at the beginning of the section, instead of jumping directly on the comment of the results

Thank you for your remark, the sentence in the previous section: "The previous figures showed the results at aerofoils level, the next phase of the analysis will focus on the integrated values." Has now been included in the section 3.13.

Also, the following has been added : " The lower RBM value in the spoiler case is explained thanks to the pitch settings, the same explanation than for the out-of-plane force FX holds for the RBM. The spoiler case pitch settings are less "aggressive" due to the higher power produced thanks to the blade inboard. The blade outboard, where most of the power is generated, is experiencing a lesser angle of attack than the no spoiler case. Therefore, the local load generated by the outer part of the blade is smaller in the no spoiler case than in the spoiler case. After the integration, performed using the equation 4, the RBMno spoiler is higher than RBMspoiler."

7. Line 215, pag. 17: authors comment the results on their approach of using the mean, the minimum and the maximum values of the polars in a quasi-steady fashion in the BEM model; however, this assumption implies to consider the rotor to behave in a quasi-steady fashion, neglecting any delly in the response or hysteretic behavior; a comment on such assumption would be recommended for making the proposed methodology more convincing

Thank you for your comment, it is indeed an implied assumption. We will modify L240 as such: "When using BEM, one cannot use a time varying description of each angle of attack during the iterative procedure. Using several steady states polars representing the different possible aerodynamic coefficients allowed for a first estimation of the load and power variation due to the unsteadiness. Analysing the loads or the different aerodynamic metrics (such as presented in Section 3.1.1 and Section 3.1.2) using three different polar states independently is acceptable because the data represents "snapshots" values. Also, in the presented results, the turbine's DoF were switched off thereby removing any hysteresis behaviour due to the blade's dynamics. Therefore, we can assume that the rotor is exhibiting a quasi-steady behaviour enabling the following comparison."

8. Line 255: a novel method is proposed to construct time series, and hence to analyze fatigue loads, by combining the oscillation in the polar with the prevalent vortex shedding frequency; while the method is very well crafted and highly appreciated, a question arises on its quantitative validity; would it be possible to explain the degree of fidelity of their technique? Ideally one could compare the results of the method with those of a high-fidelity simulation, for just one case. I understand it might not be possible in the frame of this paper, but a comment on this should be addressed in the paper.

3D CFD is the final step of the study. Unfortunately we will not have the time to include it in the present paper. It will be however a dedicated chapter in the main author's PhD thesis. At the moment we added the following to the manuscript:

"The presented method currently relies on 2D CFD simulations and BEM calculations, further studies involving 3D CFD are being carried out to assess the vortex shedding behaviour on a rotor."

9. Line 295: authors state that the results of lifetime expectancy are unrealistic, does this mean that the values in figure 18 are unreliable?

We were referring to the life expectancy reached on Figure 24 and 25. Because of the non-linear behaviour of the S-N curve, depending on the safety factor used, the life expectancy can reach up to 100 000 years. For the same reason, only a "life index" was presented and not the calculated life time.